# Epidemiology and determinants of non-diabetic hyperglycaemia and its conversion to type 2 diabetes mellitus, 2000–2015: cohort population study using UK electronic health records

Rathi Ravindrarajah [1], David Reeves [1], Elizabeth Howarth,[1] Rachel Meacock,[1] Claudia Soiland-Reyes,[2] Sarah Cotterill,[3] William Whittaker [1], Simon Heller,[4] Matt Sutton,[1] Peter Bower,[1] Evangelos Kontopantelis [1]

For numbered affiliations see end of article.

**Correspondence to**
Dr Rathi Ravindrarajah;
rathi.ravindrarajah@manchester.ac.uk

## ABSTRACT

**Objectives** To study the characteristics of UK individuals identified with non-diabetic hyperglycaemia (NDH) and their conversion rates to type 2 diabetes mellitus (T2DM) from 2000 to 2015, using the Clinical Practice Research Datalink.

**Design** Cohort study.

**Settings** UK primary Care Practices.

**Participants** Electronic health records identified 14 272 participants with NDH, from 2000 to 2015.

**Primary and secondary outcome measures** Baseline characteristics and conversion trends from NDH to T2DM were explored. Cox proportional hazards models evaluated predictors of conversion.

**Results** Crude conversion was 4% within 6 months of NDH diagnosis, 7% annually, 13% within 2 years, 17% within 3 years and 23% within 5 years. However, 1-year conversion fell from 8% in 2000 to 4% in 2014. Individuals aged 45–54 were at the highest risk of developing T2DM (HR 1.20, 95% CI 1.15 to 1.25— compared with those aged 18–44), and the risk reduced with older age. A body mass index (BMI) above 30 kg/m$^2$ was strongly associated with conversion (HR 2.02, 95% CI 1.92 to 2.13—compared with those with a normal BMI). Depression (HR 1.10, 95% CI 1.07 to 1.13), smoking (HR 1.07, 95% CI 1.03 to 1.11—compared with non-smokers) or residing in the most deprived areas (HR 1.17, 95% CI 1.11 to 1.24—compared with residents of the most affluent areas) was modestly associated with conversion.

**Conclusion** Although the rate of conversion from NDH to T2DM fell between 2010 and 2015, this is likely due to changes over time in the cut-off points for defining NDH, and more people of lower diabetes risk being diagnosed with NDH over time. People aged 45–54, smokers, depressed, with high BMI and more deprived are at increased risk of conversion to T2DM.

## Strengths and limitations of this study

► Data were based on a large, anonymised, longitudinal and nationally representative sample of general practices.
► The length of the study period (2000–2015) was useful in capturing changes over time.
► Cases of non-diabetic hyperglycaemia (NDH) and type 2 diabetes mellitus were identified using Read codes, and the quality of recording may have been problematic for the former in earlier years.
► Our NDH code list included a few relevant items and is not sensitive to misclassification.

mortality, morbidity and healthcare costs. It has been estimated that 415 million people live with diabetes across the globe and 193 million people have undiagnosed diabetes.[1] It has been suggested that currently there are 5 million people in England who are at risk of developing T2DM.[2] T2DM is characterised by pancreatic dysfunction causing insulin resistance. There are other key pathophysiological processes which increase the risk of T2DM, which involves organs including pancreas, liver, skeletal muscle, kidneys, brain, small intestine and adipose tissue.[3] Lifestyle factors such as excess weight and physical inactivity are known to increase the risk of developing T2DM.

Non-diabetic hyperglycaemia (NDH) also known as pre-diabetes or impaired glucose regulation, IGR), refers to levels of blood glucose that are increased from the normal range but not yet high enough to be in the diabetic range. Previous research has shown that individuals diagnosed with NDH are at a higher risk of developing T2DM.[4] The NHS RightCare diabetes pathway defines

## INTRODUCTION

The proportion of the population with type 2 diabetes mellitus (T2DM) has been rising globally and is an important contributor to

NDH as having an HbA1c (haemoglobin A1c or glycated haemoglobin) measurement in the 42–47 mmol/mol range (6.0%–6.4%), or fasting plasma glucose in the 5.5–6.9 mmol/mol range.[5] Previous analyses using Health Survey England data have shown discrepancies in the prevalence of NDH in the UK. While one study suggested that the average NDH prevalence was 11% in adults aged 16+ in England, in the period between 2009 and 2013,[6] the other suggested a sharp rise in the prevalence of NDH from 11.6% in 2003 to 35.3% in 2011 in all adults.[7] The use of different cut-points for HbA1C used to define NDH has been suggested as the cause of this discrepancy; one study used the National Institute for Health and Care Excellence (NICE) and Diabetes UK cut-points (HbA1C: 42–47 mmol/mol) whereas the other used the American Diabetes Association cut-points (HbA1C: 39–47 mmol/mol). Delaying or preventing T2DM has become an international priority due to the burden that the condition places on both patients and health services.[8] NHS England, Public Health England and Diabetes UK have implemented a programme to identify those at high risk of developing T2DM and offer them an evidence-based behavioural intervention (National health Service Diabetes Prevention Programme) to people identified as having NDH in an attempt to reduce the incidence of T2DM and the complications related to it.[9]

This paper explores two aspects of the epidemiology of people diagnosed with NDH in UK primary care. First, we aimed to estimate the prevalence of NDH and to explore the characteristics of patients with NDH in a population cohort of adults from 2000 to 2015. We chose this study period both to ensure high-quality data and to avoid introducing bias into our analysis from any potential effects from the National Diabetes Prevention Programme.[10] Second, we evaluated the conversion rates of NDH to T2DM over time, and whether conversion rates differ by age, sex, body mass index (BMI) levels, depression, multimorbidity and area-level deprivation.

## METHODS

### Data source

Patient-level data were obtained from the Clinical Practice Research Datalink (CPRD), one of the largest active primary care databases of electronic health records (EHR) in the UK.[11] This dataset captures approximately 7% of the total UK population. The database holds anonymised data which contains information on clinical signs, diagnoses, tests and procedures.[11] Approximately 60% of all UK CPRD practices participate in the CPRD linkage scheme, which provides additional patient-level information. For this work, we obtained patient-level deprivation through the Office of National Statistics linkage, in the form of the 2010 Index of Multiple Deprivation (IMD).[12]

### Study participants

Practices taking part in the CPRD are checked for eligibility in each year using a CPRD assessment algorithm, and evaluated to be of research standard or not. Patients were regarded as eligible if they had been registered with a practice for a full year, were aged 18 years and over and had a code for NDH between 1 April 2000 and 31 March 2016. At least one relevant Read code was considered adequate to flag a patient. Codes were identified using a strategy that involved searching for relevant terms through an algorithm, with the returned list being reviewed and finalised by members of the research team, as described elsewhere.[13 14] Read codes which were actively used by general practitioners (GPs) to identify NDH were included in the study: 44v2.00 (Glucose Tolerance Test impaired), C11y200 (Impaired glucose tolerance, IGT), C11y300 (Impaired fasting glycaemia), C11y500 (pre-diabetes), C317.00 (NDH), R102.00 ((D) Glucose Tolerance Test abnormal), R102.11 ((D) Pre-diabetes), R102.12 ((D) IGT test), R10D000 ((D) Impaired fasting glycaemia), R10D011 ((D) Impaired fasting glucose, IFG), R10E.00 ((D) IGT). Eligible patients were followed up until censored at the earliest of any of the following events: diagnosed with T2DM (the outcome event), transferred out of practice (any cause), last collection date for the practice, end date of the study (31 March 2016) or death. To report prevalence, we also included cases that were diagnosed with NDH at any point prior to 1 April 2000, who met all other inclusion criteria.

### Study measures

We calculated the prevalence of NDH in each year between 2000 and 2015, and conversion to T2DM was also determined. People with at least one relevant Read code of T2DM following the NDH diagnosis (the index date), were considered to have progressed to T2DM during the study period (online supplementary table 1 provides a list of read codes used to diagnose T2DM). Patients with a previous record of type 1 diabetes were excluded.

We extracted information on the following covariates which have previously been reported[10] to be relevant to NDH and T2DM; age, gender, BMI, total serum cholesterol, smoking status, socioeconomic status and depression. Age was grouped into the following bands: 18–44, 45–54, 55–64, 65–74, 75–84 and 85 years or over. The latest available measurement before the NDH diagnosis date, up until the previous 12 months, was used to define baseline total cholesterol and BMI. If such a value was not available, the measurement was set to missing. BMI values were classified into the following categories: underweight ($<18.5 \text{ kg/m}^2$), normal weight ($18.5–24.9 \text{ kg/m}^2$), overweight ($25.0–29.9 \text{ kg/m}^2$) and obese ($\geq30 \text{ kg/m}^2$). Total serum cholesterol in mmol/L was categorised into: under 3.0, (3.0, 4.0), (4.0, 5.0), (5.0, 6.0) and 6.0 or over. We also quantified the multimorbidity burden, using the Charlson Comorbidity Index (CCI), which is a widely used measure which assigns different weights to different conditions and includes: any malignancy, cerebrovascular disease, chronic pulmonary disease, congestive cardiac disease, dementia, HIV/AIDS, hemiplegia, lymphoproliferative disorders, metastatic solid tumour, mild liver

disease, moderate and severe liver disease (CCI also includes diabetes with complications, which we necessarily excluded).[15 16] This modified CCI was calculated using the list of validated diagnostic primary care Read codes used by Khan *et al*.[15] Participants were classified as having a condition if the condition was present at diagnosis of NDH or 12 months prior to diagnosis of NDH. CCI takes integer values and was categorised as: 0, 1–2, 3–4 and >4. Depression was evaluated using medical codes and therapy codes which were obtained from the code lists derived from the CPRD provided on a Cambridge University repository.[17] Participants were considered to have depression at the index date (the date of NDH diagnosis) if they were recorded as depressed either by a code or if they were on relevant medication in the last 12 months. Smoking status was determined from information in the patients' record and categorised as 'smoker', 'ex-smoker' or 'never smoked'. The IMD was used to classify deprivation and the IMD scores were divided into quintiles.

### Conversion of NDH to T2DM

The time of conversion of NDH to T2DM was defined as the time from the index date (diagnosis of NDH) to the date they were diagnosed as having T2DM. This time was then categorised into progression time of: 1 month; 3 months, 6 months, 12 months, 2 years, 3 years, 4 years and 5 years. Those who had a conversion time of over 5 years were excluded from analysis. In addition, patients who did not convert to T2DM, left the study or died within this study period were categorised into a single category as 'Not converted/left/died'. A small number of participants were diagnosed as having T2DM on, or ever before, the index date, and were excluded from further analyses (See figure 1).

### Statistical analysis

The characteristics of people identified with NDH are presented descriptively. Conversion rates of NDH to T2DM, in the progression time categories were plotted over time. Annual bins were defined as financial years, for example 1 April 2000 to 31 March 2001 was labelled as 2000. The associations between covariates and conversion from NDH to T2DM were estimated in a time to event analysis. A Cox proportional hazards model was employed to estimate adjusted HRs of the associations between conversion and the following covariates: gender, age groups, BMI categories, total cholesterol levels, depression, year, patient-level deprivation scores and CCI categories. Proportionality of hazards was tested using Schoenfeld residuals.

### Patient involvement

CPRD data provide anonymised patient data, hence patients are not identified by the researchers.

## RESULTS

Over the study period, a total of 148 363 participants were identified with NDH. The prevalence and incidence of NDH for each financial year is shown in table 1. Prevalence increased from 0.07% in 2000 to 1.85% in 2015. Incidence of NDH increased from 0.02% in 2000 to 0.21% in 2015. Table 2 and figure 2 show the cumulative frequency of conversion from NDH to T2DM, by year, from 1 April 2000 to 31 March 2016. Frequency of conversion within one financial year peaks in 2003 and then follows a decreasing trend. Among this general trend of declining conversion, there was a peak in the year 2011, with a further exploration of the data (results not shown) suggesting that patients had somewhat higher BMIs in this year, although that does not fully explain the rise.

After all exclusion criteria were applied (see figure 1), our final NDH population was 141 272 people, with a mean follow-up period of 5 years since the index date.

Table 3 displays the baseline characteristics of the cohort. Covariates are treated as categorical variables in our analysis, and so reported here as numbers and percentages. The mean age of the cohort was 63.2 (SD=13.4) years, and 52% were male. The prevalence of NDH was highest in those aged 65–74 years (39 178/141 272; 27.7%). The proportion of NDH was higher in older females (3728/67 369, 5.5%), compared with older males (2162/73 903; 2.9%) aged 85 years and more. The most common BMI category in our cohort was obese, with 32% of females with a measurement of BMI equal to or above 30 kg/m$^2$. Results showed that 19% of the NDH cohort had depression when they were diagnosed with NDH. The vast majority of the NDH population (85%) had a

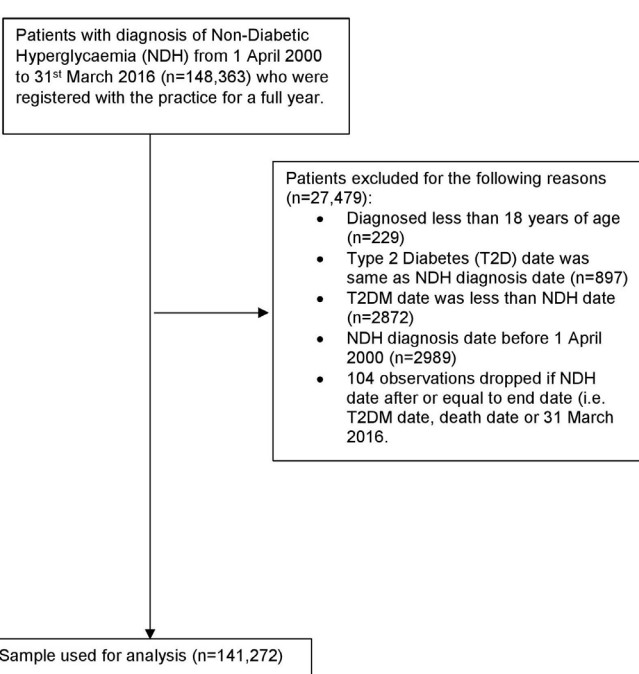

**Figure 1** Flow diagram of patient inclusions. T2DM, type 2 diabetes mellitus.

**Table 1** Prevalence and Incidence of NDH

| | Prevalence | | | Incidence | | |
|---|---|---|---|---|---|---|
| Year | Numerator | Denominator | % | Numerator | Denominator | % |
| 2000 | 2809 | 3784862 | 0.07 | 750 | 3782803 | 0.02 |
| 2001 | 4065 | 3825769 | 0.11 | 1256 | 3822960 | 0.03 |
| 2002 | 6627 | 3868575 | 0.17 | 2562 | 3864510 | 0.07 |
| 2003 | 10790 | 3905077 | 0.28 | 4163 | 3898450 | 0.11 |
| 2004 | 16687 | 3957556 | 0.42 | 5897 | 3946766 | 0.15 |
| 2005 | 23989 | 3996114 | 0.60 | 7302 | 3979427 | 0.18 |
| 2006 | 29805 | 4029795 | 0.74 | 5816 | 4005806 | 0.15 |
| 2007 | 35730 | 4074123 | 0.88 | 5925 | 4044318 | 0.15 |
| 2008 | 41930 | 4130943 | 1.02 | 6200 | 4095213 | 0.15 |
| 2009 | 48116 | 4191018 | 1.15 | 6186 | 4149088 | 0.15 |
| 2010 | 52891 | 4245410 | 1.25 | 4775 | 4197294 | 0.11 |
| 2011 | 57556 | 4283200 | 1.34 | 4665 | 4230309 | 0.11 |
| 2012 | 61787 | 4335322 | 1.43 | 4231 | 4277766 | 0.10 |
| 2013 | 68376 | 4383749 | 1.56 | 6589 | 4321962 | 0.15 |
| 2014 | 74423 | 4446718 | 1.67 | 6047 | 4378342 | 0.14 |
| 2015 | 83652 | 4528613 | 1.85 | 9229 | 4454190 | 0.21 |

Year 2000 defined as 1 April 2000 till 31 March 2001 and other years defined similarly.
NDH, non-diabetic hyperglycaemia.

Charlson comorbidity score of 0 at the index date, indicating absence of major comorbidities.

Table 4 shows the number of patients who converted from NDH to T2DM. Over the whole of the study period, the conversion rates were: 1.6% within 1 month, 3% within 3 months, 4.2% within 6 months, 7% within a year, 12.8% within 24 months, 17.2% within 3 years, 20.4% within 4 years and 22.8% over 5 years. The majority (77.2%, n=104030) did not convert, but the length of time each was followed up varied depending on the time they were diagnosed with NDH.

Table 5 shows the results from the Cox proportional hazard models, which explored time to conversion from NDH to T2DM, with failure being the diagnosis of T2DM. Residuals were linear over time, indicating that proportionality generally stood. The rate of conversion was highest for the 45–54 age group with HR 1.20 (95% CI 1.15 to 1.25), compared with those aged 18–44, and the risk steadily decreased with increasing age to an HR 0.65 (95% CI 0.60 to 0.71) for people aged 85 or over. Cholesterol categories did not appear to be strongly associated with conversion to T2DM. People with high BMI had a much higher risk of conversion to T2DM, with those classed overweight (BMI 25–30) having an HR 1.40 (95% CI 1.33 to 1.48), and those classed obese (BMI≥30) having an HR 2.0 (95% CI 1.9 to 2.1), compared with individuals with a normal BMI (18.5–25 kg/m$^2$). Compared with non-smokers, current smokers had a slightly increased risk of converting to T2DM with an HR 1.07 (95% CI of 1.03 to 1.11). Those who had a CCI score of 1–2 had a slightly higher risk of conversion to T2DM with an HR 1.1 (95% CI 1.08 to 1.15) but there was no increased risk among those with higher CCI scores. Having depression at baseline slightly increased the risk of conversion (HR 1.10, 95% CI 1.07 to 1.13). The risk of conversion to T2DM increased with patient-level deprivation as measured by the 2010 IMD, suggesting that those living in more deprived areas are at an increased risk of conversion from NDH to T2DM. Patients living in the least affluent quintile had an HR 1.17 (95% CI 1.11 to 1.24), compared with patients living in the most affluent quintile.

## DISCUSSION

In our cohort, incidence of NDH increased from 0.02% in 2000 to 0.21% in 2015. NDH is more common in males and the proportion with NDH increased with age, up to 75 years. The proportion of individuals diagnosed with NDH increased with BMI. The time taken to convert from NDH to T2DM was further explored which showed that approximately 7% converted to T2DM within a year. The conversion rates were also explored by year from 2000 to 2015, which showed a general trend of a decline in the conversion rate from NDH to T2DM with a peak in the year 2004 and 2011. The risk of conversion from NDH to T2DM was higher in men and those aged 45–54 years, decreasing with age. People with NDH who are overweight, and even more so those who are obese, have a higher risk of developing diabetes. Depression, deprivation and smoking (perhaps as a deprivation proxy) were also modestly associated with T2DM conversion.

**Table 2** Cumulative frequency of conversion from NDH to T2DM from 2000 to 2015

| Year | Within 1 month | | | | Within 3 months | | | | Within 6 months | | | | Within 1 year | | | |
|---|---|---|---|---|---|---|---|---|---|---|---|---|---|---|---|---|
| | N remaining uncon verted | N con verted | N censored | Cum % con verted | N remaining uncon verted | N con verted | N censored | Cum % con verted | N remaining uncon verted | N con verted | N censored | Cum % con verted | N remaining uncon verted | N con verted | N censored | Cum % con verted |
| 2000 | 887 | 19 | 1 | 2.1 | 870 | 13 | 4 | 3.53 | 854 | 15 | 1 | 5.2 | 818 | 25 | 11 | 7.99 |
| 2001 | 1460 | 35 | 0 | 2.34 | 1433 | 26 | 1 | 4.08 | 1397 | 29 | 7 | 6.03 | 1320 | 58 | 19 | 9.96 |
| 2002 | 2922 | 72 | 2 | 2.4 | 2863 | 55 | 4 | 4.24 | 2803 | 47 | 13 | 5.82 | 2650 | 126 | 27 | 10.07 |
| 2003 | 4793 | 115 | 5 | 2.34 | 4655 | 125 | 13 | 4.89 | 4538 | 85 | 32 | 6.63 | 4276 | 183 | 79 | 10.43 |
| 2004 | 7076 | 184 | 6 | 2.53 | 6907 | 151 | 18 | 4.62 | 6698 | 160 | 49 | 6.83 | 6370 | 241 | 87 | 10.21 |
| 2005 | 8832 | 185 | 7 | 2.05 | 8660 | 152 | 20 | 3.74 | 8479 | 132 | 49 | 5.21 | 8007 | 335 | 137 | 8.99 |
| 2006 | 8561 | 193 | 4 | 2.2 | 8389 | 149 | 23 | 3.91 | 8194 | 140 | 55 | 5.52 | 7743 | 319 | 132 | 9.23 |
| 2007 | 9240 | 192 | 14 | 2.03 | 9073 | 144 | 23 | 3.56 | 8912 | 130 | 31 | 4.95 | 8472 | 317 | 123 | 8.35 |
| 2008 | 10243 | 179 | 10 | 1.72 | 10046 | 172 | 25 | 3.37 | 9871 | 114 | 61 | 4.47 | 9391 | 370 | 110 | 8.07 |
| 2009 | 10923 | 191 | 8 | 1.72 | 10721 | 185 | 17 | 3.38 | 10553 | 123 | 45 | 4.49 | 10100 | 319 | 134 | 7.4 |
| 2010 | 9991 | 189 | 4 | 1.86 | 9828 | 146 | 17 | 3.29 | 9686 | 107 | 35 | 4.35 | 9279 | 291 | 116 | 7.24 |
| 2011 | 9973 | 163 | 6 | 1.61 | 9792 | 161 | 20 | 3.2 | 9628 | 126 | 38 | 4.45 | 9181 | 309 | 138 | 7.53 |
| 2012 | 10057 | 162 | 5 | 1.58 | 9912 | 130 | 15 | 2.86 | 9743 | 131 | 38 | 4.14 | 9366 | 274 | 103 | 6.85 |
| 2013 | 12267 | 131 | 17 | 1.06 | 12130 | 110 | 27 | 1.94 | 11963 | 115 | 52 | 2.88 | 11537 | 264 | 162 | 5.03 |
| 2014 | 11318 | 85 | 14 | 0.74 | 11214 | 71 | 33 | 1.37 | 11061 | 92 | 61 | 2.18 | 10717 | 209 | 135 | 4.04 |
| 2015 | 12832 | 81 | 1080 | 0.6 | 10111 | 85 | 2636 | 1.34 | 6716 | 72 | 3323 | 2.18 | 0 | 137 | 6566 | 27.32 |

| Year | Within 2 years | | | | Within 3 years | | | | Within 4 years | | | | Within 5 years | | | |
|---|---|---|---|---|---|---|---|---|---|---|---|---|---|---|---|---|
| | N remaining uncon verted | N con verted | N censored | Cum % con verted | N remaining uncon verted | N con verted | N censored | Cum % con verted | N remaining uncon verted | N con verted | N censored | Cum % con verted | N remaining uncon verted | N con verted | N censored | Cum % con verted |
| 2000 | 734 | 62 | 22 | 15.06 | 634 | 68 | 32 | 23.1 | 545 | 57 | 32 | 30.2 | 456 | 60 | 29 | 38.09 |
| 2001 | 1160 | 103 | 57 | 17.14 | 971 | 135 | 54 | 27.01 | 827 | 94 | 50 | 34.26 | 694 | 76 | 57 | 40.52 |
| 2002 | 2283 | 256 | 111 | 18.95 | 1973 | 210 | 100 | 26.57 | 1674 | 198 | 101 | 34.13 | 1377 | 191 | 106 | 41.89 |
| 2003 | 3647 | 437 | 192 | 19.8 | 3105 | 359 | 183 | 27.89 | 2672 | 272 | 161 | 34.38 | 2305 | 228 | 139 | 40.13 |
| 2004 | 5490 | 590 | 290 | 18.72 | 4726 | 471 | 293 | 25.88 | 4086 | 384 | 256 | 32.07 | 3533 | 325 | 228 | 37.63 |
| 2005 | 6939 | 711 | 357 | 17.25 | 6025 | 577 | 337 | 24.3 | 5275 | 459 | 291 | 30.21 | 4650 | 406 | 219 | 35.7 |
| 2006 | 6741 | 700 | 302 | 17.6 | 5841 | 638 | 262 | 25.55 | 5076 | 467 | 298 | 31.66 | 4468 | 341 | 267 | 36.37 |
| 2007 | 7328 | 829 | 315 | 17.49 | 6385 | 643 | 300 | 24.88 | 5612 | 484 | 289 | 30.71 | 4959 | 379 | 274 | 35.5 |
| 2008 | 8176 | 836 | 379 | 16.42 | 7247 | 602 | 327 | 22.7 | 6473 | 474 | 300 | 27.86 | 5763 | 421 | 289 | 32.66 |
| 2009 | 9059 | 708 | 333 | 14 | 8049 | 621 | 389 | 20.02 | 7229 | 500 | 320 | 25.09 | 6597 | 344 | 288 | 28.73 |
| 2010 | 8324 | 616 | 339 | 13.51 | 7427 | 587 | 310 | 19.73 | 6712 | 440 | 275 | 24.57 | 6186 | 306 | 220 | 28.07 |
| 2011 | 8091 | 773 | 317 | 15.46 | 7303 | 473 | 315 | 20.5 | 6703 | 342 | 258 | 24.29 | 0 | 137 | 6566 | 27.32 |
| 2012 | 8467 | 537 | 362 | 12.30 | 7769 | 366 | 332 | 16.17 | | | | | | | | |
| 2013 | 10625 | 487 | 425 | 9.12 | | | | | | | | | | | | |

NDH, non-diabetic hyperglycaemia; T2DM, type 2 diabetes mellitus.

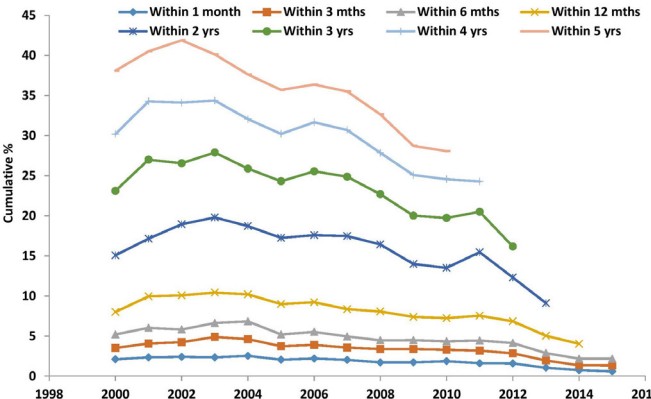

**Figure 2** Cumulative conversion of NDH to T2DM diabetes from 2000 to 2015. Year 2000 defined as 1 April 2000 till 31 st March 2001 and other years defined similarly. NDH, non-diabetic hyperglycaemia; T2DM, type 2 diabetes mellitus.

Our study has several strengths. It was based on a large, longitudinal and nationally representative data resource. The length of the study period is also useful in capturing changes over time. This study has some limitations. Our diagnosed cases of NDH and T2DM are based on Read codes being used. Although we could have considered other approaches to define NDH and T2DM to avoid false positives, in the context of the UK primary care, coding of T2DM is known to be of very high quality because of the Quality and Outcomes Framework (QOF), which incentive GPs for accurate recording.[14] Although this change occurred in 2004, quality was already high from 2000 onwards, in anticipation for the scheme and other smaller-scale frameworks. The only potential issue with the QOF was the non-distinction in coding between type 1 and type 2, until explicitly requested in 2006. This may have led to us missing a few cases that exited the database before 2006, if they had type 2 diabetes but were only given a generic diabetes code. In our experience this is very rare, however, and it would not affect our finding that conversion rates for NDH have dropped over time. As previously mentioned, the quality of recording is very high and people associated with a Read code for T2DM, have the condition—there is no provisional coding and GPs are encouraged to add to records only if certain since they know retracting such a diagnosis is very complicated. If someone is suspected of having the condition they will be not be given a Read code, but information will be added in notes (or with a 'suspected diabetes' code). Remission is possible of course, although rare, but it is not relevant for this study (where T2DM is the outcome of interest in a time to event analysis).

Regarding NDH coding, the situation is more complicated because of the absence of financial incentives through the QOF, hence practice variability is greater. In addition, the definition of NDH has changed over time, as we explain in the paper, making it difficult to operationalise through biological measurements, which are very often missing.

Estimates from EHRs are sensitive to the code lists and that our findings need to be interpreted with caution,[18] however, our code list included only a few relevant items and is not sensitive to misclassification. For BMI and cholesterol, we categorise and include a 'missing' category, which can be problematic, but allows us to observe the associations with T2DM conversion. Our risk prediction model did not attempt to include and reaffirm all known drivers of diabetes, but we primarily aimed to examine the role of socioeconomic drivers and lifestyle factors, along with depression (potentially actionable and important comorbidity for T2DM,[19] and a proxy for 'overall health'. Alcohol intake was not included in the model, since the quality of recording such information in UK primary care is rather poor.[20] We also decided not to use medication for two reasons: first, we would need to capture and organise everything to a patient (and the relevant volumes), which is a tremendous amount of work, with no clear link to conversion as far as we know; second, and more importantly, including treatment in our model would probably introduce unmeasured confounding, with treatments being associated to conversion when the underlying conditions and the health of the patient are the driving causes.

Our findings suggested the women were at a lower risk of conversion from NDH to T2DM than men. Previous studies have shown that the incidence of diabetes in those diagnosed with pre-diabetes was higher in women.[10] The difference may be due to different populations studied (two of the three studies were on American Indians and the other was an Australian population). The discrepancies may also be due to the different definition of NDH used.[21] For example in the Australian study which followed up 5842 participants over 5 years, men categorised as having IFG had a higher incidence of diabetes compared with women (4.0% vs 2.0%), whereas women categorised with IGT had a significantly higher incidence of diabetes than men (4.4% vs 2.9%).[22]

A review[23] exploring the rates of conversion from IGT to T2DM showed rates ranging from 1.5% per year in Bradford, UK to 7% in Mexicans and Americans. In our study, rates of conversion from NDH to T2DM decreased from 2000 to 2015, with peaks in 2004 and 2011. Since studies in primary care data have suggested that the incidence rates of T2DM has stabilised after 2005,[24] this apparent decrease in conversion rates needs to be interpreted with caution. One possible explanation is changes in the definition of NDH, with different HbA1c ranges used over the study period. Another plausible explanation for the decreasing trends is changes in coding practice, with more people of lower conversion risk being linked with NDH in primary care records. In addition, the peak we observed for 2011 might either be due to the uptake of NHS Health Checks which was introduced in April 2009 and also WHO recommendation in 2011 to use HbA1c for T2DM diagnosis.[25] A systematic review exploring the trends of pre-diabetes in South Asians, showed that T2DM was rising but the prevalence of IGT was stable

**Table 3** Characteristics of the cohort

|  | All | Males | Females |
|---|---|---|---|
| N | 141272 | 73903 (52.3) | 67369 (47.7) |
| Age (years) | 63.2±13.4 | 62.8±12.4 | 63.6±14.5 |
| Age group count (%) |  |  |  |
| 18–44 | 12896 (9.1) | 5619 (7.6) | 7277 (10.8) |
| 45–54 | 22717 (16.1) | 12934 (17.5) | 9783 (14.5) |
| 55–64 | 36790 (26.0) | 21127 (28.6) | 15663 (23.3) |
| 65–74 | 39178 (27.7) | 21042 (28.5) | 18136 (26.9) |
| 75–84 | 23801 (16.9) | 11019 (14.9) | 12782 (19.0) |
| ≥85 | 5890 (4.2) | 2162 (2.9) | 3728 (5.5) |
| Smoking status count (%) |  |  |  |
| Current | 21088 (14.9) | 11352 (15.4) | 9736 (14.5) |
| Ex | 46301 (32.8) | 27979 (37.9) | 18322 (27.2) |
| Never | 27834 (19.7) | 12046 (16.3) | 15788 (23.4) |
| Missing | 46049 (32.6) | 22526 (30.5) | 23523 (34.9) |
| BMI categories (kg/m$^2$) count (%) |  |  |  |
| <18.5 | 628 (0.4) | 153 (0.2) | 475 (0.7) |
| 18.5–25 | 11553 (8.2) | 5504 (7.5) | 6049 (9.0) |
| 25–30 | 27523 (19.5) | 16686 (22.6) | 10837 (16.1) |
| ≥30 | 42456 (30.1) | 21189 (28.7) | 21267 (31.6) |
| Missing | 59112 (41.8) | 30371 (41.1) | 28741 (42.7) |
| Cholesterol (%) count (%) |  |  |  |
| <3 | 1538 (1.1) | 1203 (1.6) | 336 (0.5) |
| 3–4 | 12668 (9.0) | 8814 (11.9) | 3859 (5.7) |
| 4–5 | 29204 (20.7) | 17170 (23.2) | 12041 (17.9) |
| 5–6 | 28554 (20.2) | 14889 (20.1) | 13670 (20.3) |
| ≥6 | 22818 (16.2) | 9844 (13.3) | 12979 (19.3) |
| Missing | 46490 (32.9) | 22002 (29.8) | 24513 (36.4) |
| Depression | 26064 (18.5) | 9724 (13.2) | 16340 (24.3) |
| CCI score count (%) |  |  |  |
| None | 120158 (85.1) | 63571 (86.0) | 56587 (84.0) |
| 1–2 | 20912 (14.8) | 10215 (13.8) | 10697 (15.9) |
| 3–4 | 142 (0.1) | 85 (0.1) | 57 (0.1) |
| >4 | 60 (0.04) | 32 (0.04) | 28 (0.04) |
| Patient-level deprivation index (2010 IMD score) count (%) |  |  |  |
| Quintile 1(most affluent) | 12854 (9.1) | 7034 (9.5) | 5820 (8.6) |
| Quintile 2 | 13617 (9.6) | 7368 (10.0) | 6249 (9.3) |
| Quintile 3 | 12882 (9.1) | 6692 (9.1) | 6190 (9.2) |
| Quintile 4 | 12816 (9.1) | 6514 (8.8) | 6302 (9.4) |
| Quintile 5 (least affluent) | 9866 (7.0) | 4780 (6.5) | 5086 (7.6) |
| Missing | 79237 (56.1) | 41515 (56.2) | 37722 (56.0) |

BMI, body mass index; CCI, Charlson Comorbidity Index; IMD, Index of Multiple Deprivation.

or decreasing. They suggested that this might be due to increased testing for T2DM and also studies have found that fasting plasma glucose was more influenced by obesity than 2-hour glucose testing.[26] It has also been suggested that these decreased trends might be due to a more rapid progression from IGT to T2DM with the IGT state possibly skipping altogether in the disease progression.[27] Studies have also shown a change of NDH to normoglycaemia after lifestyle and drug-based interventions, which might also be a reason for our findings,[28 29] as the NICE

**Table 4** Conversion from at risk of diabetes (NDH) to T2DM

| Time taken to convert from at risk to type 2 diabetes (T2D) | Numerator (total number diagnosed with T2D) | Denominator (total number with NDH) | % | % Change |
|---|---|---|---|---|
| Within 1 month | 2176 | 134 734 | 1.62 | |
| Within 3 months | 4051 | 134 734 | 3.01 | 1.39 |
| Within 6 months | 5669 | 134 734 | 4.21 | 1.20 |
| Within 1 year | 9369 | 134 734 | 6.95 | 2.75 |
| Within 2 years | 17 216 | 134 734 | 12.78 | 5.82 |
| Within 3 years | 23 168 | 134 734 | 17.20 | 4.42 |
| Within 4 years | 27 490 | 134 734 | 20.40 | 3.21 |
| Within 5 years | 30 704 | 134 734 | 22.79 | 2.39 |

NDH, non-diabetic hyperglycaemia; T2DM, type 2 diabetes mellitus.

guidelines have also proposed primary care practitioners to advice patients with NDH on diet and exercise as well as drug interventions with metformin in some cases.[30] We found a crude rate of conversion of NDH to T2DM to be about 7%, where a previous report using CPRD in which pre-diabetes was defined using Fasting glucose levels showed the progression of IFG to diabetes was 6% per year.[31]

The prevalence of NDH in Health Survey England analyses showed an increase with age, and it increased from 3% in 16–69 age groups to 30.4% in those aged over 80 years.[10] However, our findings showed the risk of conversion to diabetes from NDH decreased with increasing age and the risk was significantly lower in those aged over 75 years compared with those aged 18–44. Similar associations were shown in The Strong Heart Study which suggested that this might be due to the survival effect in the older adults and the prevalence of obesity being higher in younger adults.[32] An analysis of six prospective studies which explored the predictors of progression from IGT to non-insulin dependent diabetes mellitus (NIDDM) found inconsistent relationships with age. In the studies with the highest incidence rates of NIDDM, the progression of NIDDM increased with age in participants diagnosed with IGT at a younger age and decreased with age in participants who were diagnosed with IGT at an older age.[33] There was a negative association in those aged over 85 years and the risk of conversion from NDH to T2DM. This negative association may be due to the fact older population may be less likely to be checked for T2DM in primary care[31] or the threshold needed to identify NDH in older adults may need to be reconsidered.

We also found the risk of conversion of NDH increased with increase in BMI. Obesity has been linked to increased prevalence of pre-diabetes previously,[34] however, several other studies exploring the progression of pre-diabetes to T2DM have shown conflicting results with BMI playing a small or non-significant role.[33]

We also showed that current smokers were more likely to convert from NDH to T2DM. In the Health Survey England data, it was shown that the prevalence of pre-diabetes was significantly higher in ex-smokers compared with non-smokers.[10] Our results also showed a high cholesterol levels were associated with a reduced risk of developing T2DM. Previous studies to our knowledge have not explored the relation of cholesterol with progression of pre-diabetes to diabetes. Our findings also indicated that having a 1–2 Charlson comorbidity score increased the risk of progression to T2DM; however, we were not able to distinguish which comorbidities were linked to progression from NDH to T2DM.

Socioeconomic inequalities exist in healthcare, a fact that has been summarised by Hart's inverse care law which suggests that those in most need of healthcare are those least likely to receive it.[35] Our findings that the risk of conversion of NDH to T2DM was higher in those of lower socioeconomic status has not been reported previously, to our knowledge. Although a previous report on NDH by Public Health England using the Health Survey England data showed that there was no significant difference in the prevalence of NDH by quintile of deprivation, the study did not explore the risk of conversion from NDH to T2DM.[10] Our results align with previous findings which have suggested that IGR/NDH and T2DM are more prevalent in those with low socioeconomic status.[6 7]

## CONCLUSIONS

Over the study period, the conversion rate of NDH to T2DM was, on average, 7% within a year. However, there was a large reduction in that rate over time, which should be attributed to changes in coding practices and in the definition of NDH, rather than a reduction in the incidence of T2DM. The key predictors in the progression of NDH to T2DM were age, increased BMI and lower socioeconomic status. It would be interesting to examine the population trends of progression from NDH to T2DM following the introduction of the National Diabetes Prevention Programme, a behavioural intervention

**Table 5** Cox proportional hazard models exploring time to conversion from NDH to T2DM for patients by baseline characteristics

| | HR (95% CI) | P value |
|---|---|---|
| Males | Ref | |
| Females | 0.97 (0.95 to 0.99) | 0.009 |
| Age group (years) | | |
| 18–44 | Ref | |
| 45–54 | 1.20 (1.15 to 1.25) | <0.001 |
| 55–64 | 1.10 (1.06 to 1.14) | <0.001 |
| 65–74 | 1.03 (0.99 to 1.07) | 0.13 |
| 75–84 | 0.86 (0.82 to 0.90) | <0.001 |
| ≥85 | 0.65 (0.60 to 0.71) | <0.001 |
| Cholesterol categories (%) | | |
| <3 | 1.04 (0.95 to 1.16) | 0.391 |
| 3–4 | 1.03 (0.99 to 1.07) | 0.165 |
| 4–5 | Ref | |
| 5–6 | 0.94 (0.92 to 0.98) | 0.001 |
| ≥6 | 0.92 (0.89 to 0.95) | <0.001 |
| Missing | 0.91 (0.89 to 0.94) | <0.001 |
| Smoking status | | |
| Non smoker | Ref | |
| Current smoker | 1.07 (1.03 to 1.11) | <0.001 |
| Ex- smoker | 0.98 (0.96 to 1.01) | 0.312 |
| missing | 0.98 (0.95 to 1.02) | 0.338 |
| BMI categories (kg/m$^2$) | | |
| <18.5 | 0.59 (0.44 to 0.78) | <0.001 |
| 18.5–25 | Ref | |
| 25–30 | 1.40 (1.33 to 1.48) | <0.001 |
| ≥30 | 2.02 (1.92 to 2.13) | <0.001 |
| Missing | 1.44 (1.37 to 1.52) | <0.001 |
| Depression | 1.10 (1.07 to 1.13) | <0.001 |
| CCI Score | | |
| None | Ref | |
| 1–2 | 1.11 (1.08 to 1.15) | <0.001 |
| 3–4 | 0.98 (0.68 to 1.43) | 0.934 |
| >4 | 1.67 (0.99 to 2.81) | 0.057 |
| Patient-level deprivation index | | |
| Quintile 1(most affluent) | Ref | |
| Quintile 2 | 1.08 (1.03 to 1.13) | 0.002 |
| Quintile 3 | 1.03 (0.98 to 1.08) | 0.237 |
| Quintile 4 | 1.12 (1.07 to 1.18) | <0.001 |
| Quintile 5 (least affluent) | 1.17 (1.11 to 1.24) | <0.001 |
| Missing | 1.13 (1.09 to 1.18) | <0.001 |
| Year trend | 0.94 (0.94 to 0.95) | <0.001 |

BMI, body mass index; CCI, Charlson Comorbidity Index; NDH, non-diabetic hyperglycaemia; T2DM, type 2 diabetes mellitus.

programme targeted at people with a high risk of developing T2DM.[9]

**Author affiliations**
[1]Division of Population Health,Faculty of Biology, Medicine and Health, The University of Manchester, Manchester, UK
[2]Research & Innovation, Northern Care Alliance NHS Group, Summerfield House, M5 5AP, Salford, UK
[3]Centre for Biostatistics, University of Manchester, Manchester, UK
[4]Academic Unit of Diabetes, Endo and Metab, Unversity of Sheffield, Sheffield, UK

**Contributors** EK and RR designed the study, RR extracted the data from all sources and performed the analyses. RR wrote the manuscript. DR, EH, RM, CS-R, SC, WW, SH, MS, PB and EK critically revised the manuscript. RR is the guarantor of this work and, as such, had full access to all the data in the study and takes responsibility for the integrity of the data and the accuracy of the data analysis.

**Funding** This manuscript is independent research funded by the National Institute for Health Research (Health Services and Delivery Research, 16/48/07 – Evaluating the NHS Diabetes Prevention Programme (NHS DPP): the DIPLOMA research programme (Diabetes Prevention – Long Term Multimethod Assessment)).

**Competing interests** National Institute for Health Research (Health Services and Delivery Research, 16/48/07 – Evaluating the NHS Diabetes Prevention Programme (NHS DPP): the DIPLOMA research programme (Diabetes Prevention – Long Term Multimethod Assessment)). Funded the time and facilities of RR. SH contributes for consultancy for Eli Lilly, NovoNordisk, Takeda, Sanofi Aventis, Zealand Pharma, UN-EEG and is also part of the speakers panel for NovoNordisk.

**Patient and public involvement** Patients and/or the public were not involved in the design, or conduct, or reporting, or dissemination plans of this research.

**Patient consent for publication** Not required.

**Ethics approval** The protocol for this study received scientific and ethical approval from the Independent Scientific Advisory Committee for CPRD studies (ISAC Protocol 18_101).

**Provenance and peer review** Not commissioned; externally peer reviewed.

**Data availability statement** No data are available. The data used in this study cannot be shared due to licensing restrictions by CPRD.

**ORCID iDs**
Rathi Ravindrarajah http://orcid.org/0000-0003-4875-4912
David Reeves http://orcid.org/0000-0001-6377-6859
William Whittaker http://orcid.org/0000-0003-2530-0360
Evangelos Kontopantelis http://orcid.org/0000-0001-6450-5815

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
