## [Reviewer comments · BMJ Open]

ARTICLE DETAILS

TITLE (PROVISIONAL)	The epidemiology and determinants of non-diabetic hyperglycaemia and its conversion to type 2 diabetes mellitus, 2000-2015: cohort population study using UK electronic health records.
AUTHORS	Ravindrarajah, Rathi; Reeves, David; Howarth, Elizabeth; Meacock, Rachel; Soiland-Reyes, Claudia; Cotterill, Sarah; Whittaker, William; Heller, Simon; Sutton, Matt; Bower, Peter; Kontopantelis, Evangelos

VERSION 1 – REVIEW

REVIEWER	You Chen Vanderbilt University Medical Center
REVIEW RETURNED	26-Sep-2019

GENERAL COMMENTS	Studying characteristics of non-diabetic hyperglycemia (NDH) patients and determining factors leading the conversion from NDH to type 2 diabetes mellitus (T2DM) are very important research objectives. The authors investigated EHRs of 14,272 patients with NDH, ranging from 2000 to 2015. They found patients aged 45-54, BMI above 30, depression, smoking, or residing in the most deprived areas were associated with conversion. It is a great work, and the paper is well written. The reviewer has the following several concerns. It is unclear if the authors used identified patient information. They should explicitly state identified or de-identified patient information they used. According to the reviewer's understanding, authors used identified information. Otherwise it is impossible to get patient-level deprivation data. Using only read codes to build cohorts of NDH and T2DM is insufficient. Only relying on codes to identify NDH and T2DM will have a high false-positive rate – most of patients without NDH/T2DM may be identified as NDH/T2DM. There are several T2DM phenotyping algorithms in the literature. Authors should investigate these algorithms and determine if they can use the algorithms in their study. It seems almost all factors identified to be related to the conversion from NDH to T2DM are already known. It would be better if authors can list factors already known and those which are novel in a table. Otherwise, it is hard to determine what is the contribution of the work.
---

REVIEWER	Joel Dave Groote Schuur Hospital and University of Cape Town
REVIEW RETURNED	20-Dec-2019

GENERAL COMMENTS	Please maintain consistency in the abbreviation of type 2 diabetes mellitus, choose either T2DM or T2D or T2diabetes (all 3 have been used in the manuscript) I would recommend that the authors comment on which Read code was most commonly used to detect NDH and whether NDH defined from either a fasting glucose or an abnormal glucose tolerance test was more predictive of converting to T2DM page 2, line 27 and page 7, line 50 - please add units after BMI value of 30 page 4, line 4 - please review the concept that pancreatic dysfunction causes insulin resistance in patients with T2DM. Pancreatic dysfunction and insulin resistance are 2 of at least 8 pathophysiological processes increasing risk of T2DM. There are multiple potential causes of insulin resistance. page 4, line 26 - is the the range 5.5-6.9 mmol/L correct? Most international societies define abnormal glucose as a fasting level of 5.6 mmol/L or greater. A glucose value of 5.5 mmol/L would generally be considered normal page 6, line 8 - why was alcohol intake, previous pancreatitis and current medication not included as relevant for NDH and conversion of NDH to T2DM? If this was not available then it should be added as a limitation of the study page 6, line 55 - why were those that converted to T2DM after 5 years excluded from the analysis? Page 17 - please add BMI units to Table 3 Page 19 - please add units to Table 5 page 23, line 10 - was ethics approval given for this study? If it wasn't obtained then can the authors please provide an explanation. At the end of the study under "Ethical Approval" it states "Not applicable". For all studies it is applicable and should be obtained, especially if the study it is to be published.
--

VERSION 1 – AUTHOR RESPONSE

Reviewer: 1

Reviewer Name: You Chen

Institution and Country: Vanderbilt University Medical Center

Please state any competing interests or state 'None declared': I have no competing interests.

Please leave your comments for the authors below

Studying characteristics of non-diabetic hyperglycemia (NDH) patients and determining factors leading the conversion from NDH to type 2 diabetes mellitus (T2DM) are very important research objectives. The authors investigated EHRs of 14,272 patients with NDH, ranging from 2000 to 2015. They found patients aged 45-54, BMI above 30, depression, smoking, or residing in the most deprived areas were associated with conversion. It is a great work, and the paper is well written. The reviewer has the following several concerns.

It is unclear if the authors used identified patient information. They should explicitly state identified or

de-identified patient information they used. According to the reviewer's understanding, authors used identified information. Otherwise it is impossible to get patient-level deprivation data.

Using only read codes to build cohorts of NDH and T2DM is insufficient.

Response:

Thank you for your helpful comments.

Regarding whether the data was identified or de-identified we used data from the Clinical Practice Research Data-link which only provides anonymised health data to researchers. We agree that this was not made clear hence have added this information on the Strengths and limitations sections (Page 3, line 2) and also in the methods (Page 5, line 5-6)

Only relying on codes to identify NDH and T2DM will have a high false-positive rate – most of patients without NDH/T2DM may be identified as NDH/T2DM. There are several T2DM phenotyping algorithms in the literature. Authors should investigate these algorithms and determine if they can use the algorithms in their study.

Response:

This is a useful suggestion and indeed we have considered alternative approaches to define NDH and T2DM. We are aware of the various T2DM algorithms that are available, which are focused on the whole population, rather than people diagnosed with NDH. In the context of the UK primary care, coding of T2DM is known to be of very high quality because of the Quality and Outcomes Framework (QOF), which incentive GPs for various activities, the prerequisite for which was accurate recording (e.g. see <https://www.ncbi.nlm.nih.gov/pmc/articles/PMC4769302/>). Although this change occurred in 2004, quality was already high from 2000 onwards, in anticipation for the scheme and other smaller-scale frameworks. The only potential issue with the QOF was the non-distinction in coding between Type-1 and Type-2, until explicitly requested in 2006. This may have led to us missing a few cases that exited the database before 2006 (at which point it time they would have to be given a specific code to be included in the QOF returns), if they had type-2 diabetes but were only given a generic diabetes code. In our experience this is very rare, however and it would not affect our finding that conversion rates for NDH have dropped over time. In terms of false-positive rates, in the past we have experimented with defining cohorts of chronic conditions differently (i.e. medications and two or more relevant Read codes), but we found that resulting changes were negligible. As previously mentioned, the quality of recording is very high and people associated with a Read code for T2DM, have the condition – there is no provisional coding and GPs are encouraged to add to records only if certain since they know retracting such a diagnosis is very complicated. If someone is suspected of having the condition they will be not be given a Read code, but information will be added in notes (or with a “suspected diabetes” code). Remission is possible of course, although rare, but it is not relevant for this study (where T2DM is the outcome of interest in a time to event analysis). Regarding NDH coding, the situation is more complicated because of the absence of financial incentives through the QOF, hence practice variability is greater. In addition, the definition of NDH has changed over time, as we explain in the paper, making it difficult to operationalise through biological measurements, which are very often missing. As for T2DM, however, we would expect many cases of false positives, as the reviewers suggests, because of the coding practices previously explained. We would expect many cases to be missed, something that is well known and acknowledged in the paper, but something that should have no significant bearing on our findings and their implications (unless there is something fundamentally different about the “missed” NDH cases, which we do not think is the case).

It seems almost all factors identified to be related to the conversion from NDH to T2DM are already known. It would be better if authors can list factors already known and those which are novel in a table. Otherwise, it is hard to determine what is the contribution of the work.

Response:

Thank you for this suggestion. As the literature has shown discrepancies in the factors related to the conversion from NDH to T2D, our results try to explain these discrepancies and also the lack of clear and consistent definition for the term NDH. In addition, we reported that women had a lower risk of conversion from NDH to T2D than men. This is different to previous studies in the literature (Page 9 lines 7-15). We have also reported that high cholesterol levels were associated with a reduced risk of developing T2DM (Page 10, lines 28-29), which has not been previously reported in the literature. We discuss all predictors in the paper, in relation to previous work, but we feel that organising in a table based on novelty is problematic and subjective and we would prefer to do so in text.

Reviewer: 2

Reviewer Name: Joel Dave

Institution and Country: Groote Schuur Hospital and University of Cape Town

Please state any competing interests or state 'None declared': None declared

Please leave your comments for the authors below

Please maintain consistency in the abbreviation of type 2 diabetes mellitus, choose either T2DM or T2D or T2diabetes (all 3 have been used in the manuscript)

Response:

Thank you for noticing this error. It has now been changed throughout the document to type 2 diabetes mellitus with the T2DM abbreviation being used.

Page 9, line 7; Page 10, line 26, 31 & 33: Page 11, line 7: Page 13,14,16,18

I would recommend that the authors comment on which Read code was most commonly used to detect NDH and whether NDH defined from either a fasting glucose or an abnormal glucose tolerance test was more predictive of converting to T2DM

Response:

Thank you for this suggestion. This is something we considered as well but we decided against reporting for a couple of reasons. First, it is secondary to the aims of the paper and the paper is already quite complicated and long. Second, Read code usage changes over time and is often computer system specific (so may not be generalisable to England/the UK), hence this is as simple a task as it may originally seem, while its usefulness is perhaps questionable.

In terms of how NDH is defined, we did conduct secondary analyses for 5 different cohorts, according to the different definitions of the NDH in the literature exploring the Fasting Plasma glucose tolerance tests (FPG) as well as abnormal glucose tolerance test (HBA1C categories). The categories of NDA definitions we explored were

1) American Diabetes Association (ADA) FPG (5.6-6.9)mmol/mol

2) ADA HBA1C (39-46)mmol/mol (This was also explored further by categorising HBA1C levels to quartiles)

3) WHO FPG (6.1-6.9)mmol/L

4) International Expert Committee [IEC] HBA1C (42-46)mmol/mol

5) Diabetes UK {FBG(5.5-6.9) orHBA1C (42-47)}

We explored whether conversion to T2DM varied across these definitions. Although some variability was observed it did not explain the drop in conversion rates over time, which one of the key findings of the study. Thus, we concluded that this analysis would not add much to the paper (considering the significant expansion needed to explain the cohorts and the methods), and was not included.

Below are the plotted patterns for these cohorts

Figure 1: Cumulative conversion of NDH (HBA1C (39-46)mmol/mol) to T2DM from 2000 till 2015

Figure 2: Cumulative conversion of NDH (American Diabetes Association (ADA) FPG (5.6-6.9)mmol/mol) to T2DM from 2000 till 2015

Figure 3: Cumulative conversion Diabetes UK {FBG (5.5-6.9) or HbA1C (42-47)}to T2DM from 2000 till 2015

Figure 4: Cumulative conversion of NDH (WHO FPG (6.1-6.9)mmol/L) to T2DM from 2000 till 2015

Figure 5: Cumulative conversion of NDH IEC {HBA1C (42-46) mmol/mol} to T2DM from 2000 till 2015

page 2, line 27 and page 7, line 50 - please add units after BMI value of 30

Response:

Corrected, thank you

page 4, line 4 - please review the concept that pancreatic dysfunction causes insulin resistance in patients with T2DM. Pancreatic dysfunction and insulin resistance are 2 of at least 8 pathophysiological processes increasing risk of T2DM. There are multiple potential causes of insulin resistance.

Response:

This has been reviewed and a sentence has been added on Page 4, line 7-9.

“There are other key pathophysiological processes which increase the risk of T2DM, which involve organs including pancreas, liver, skeletal muscle, kidneys, brain, small intestine and adipose tissue³”.

page 4, line 26 - is the the range 5.5-6.9 mmol/L correct? Most international societies define abnormal glucose as a fasting level of 5.6 mmol/L or greater. A glucose value of 5.5 mmol/L would generally be considered normal

Response:

We have checked this again and can confirm this are the guidelines provided by the NHS to be referred into the NHS Diabetes Prevention Programme, please find attached link to the document. The reference is provided in page 2.

<https://www.england.nhs.uk/rightcare/wp-content/uploads/sites/40/2018/07/nhs-rightcare-pathway-diabetes.pdf>

page 6, line 8 - why was alcohol intake, previous pancreatitis and current medication not included as relevant for NDH and conversion of NDH to T2DM? If this was not available then it should be added as a limitation of the study

Response:

Thank you, these are good points. Quality of recording for alcohol is poor in the database, as it is in UK primary care in general, and we decided against using. We also decided not to use medication for two reasons: first, we would need to capture and organise everything to a

patient (and the relevant volumes), which is a tremendous amount of work, with no clear link to conversion as far as we know; secondly, and more importantly, including treatment in our model would probably introduce unmeasured confounding, with treatments being associated to conversion when the underlying conditions and the health of the patient are the driving causes. Regarding pancreatitis, this was an omission and we thank again the reviewer for highlighting this.

Our risk prediction model did not attempt to include and reaffirm all known drivers of diabetes, but we primarily aimed to examine the role of socio-economic drivers and lifestyle factors, along with “overall health” (using the Charlson Comorbidity Index as a proxy), and depression which has been found to be particularly important (and potentially actionable) in the context of T2DM. We have now expanded the limitations section to explain our modelling choices.

Page 9 , lines 9-18

“Our risk prediction model did not attempt to include and reaffirm all known drivers of diabetes, but we primarily aimed to examine the role of socio-economic drivers and lifestyle factors, along with depression (potentially actionable and important comorbidity for T2DM ¹⁹), and a proxy for “overall health”. Alcohol intake was not included in the model, since the quality of recording such information in UK primary care is rather poor ²⁰. We also decided not to use medication for two reasons: first, we would need to capture and organise everything to a patient (and the relevant volumes), which is a tremendous amount of work, with no clear link to conversion as far as we know; secondly, and more importantly, including treatment in our model would probably introduce unmeasured confounding, with treatments being associated to conversion when the underlying conditions and the health of the patient are the driving causes. “

page 6, line 55 - why were those that converted to T2DM after 5 years excluded from the analysis?

Response:

As the number of years taken to convert from NDH to T2DM ranged up to 22 years where the numbers converted at the extreme years were quite low, for the consistency in our analysis we decided to restrict our analysis to those who converted from NDH to T2DM within a 5 year period.

Page 17 - please add BMI units to Table 3

Page 19 - please add units to Table 5

Response:

Amended, thanks

page 23, line 10 - was ethics approval given for this study? If it wasn't obtained then can the authors please provide an explanation. At the end of the study under "Ethical Approval" it states "Not applicable". For all studies it is applicable and should be obtained, especially if the study it is to be published.

Response:

This has now been amended. Page 23, lines (8-9)

“The protocol for this study received scientific and ethical approval from the Independent Scientific Advisory Committee for CPRD studies (ISAC Protocol 18_101).”